# Are Fear of COVID-19 and Vaccine Hesitancy Associated with COVID-19 Vaccine Uptake? A Population-Based Online Survey in Nigeria

**DOI:** 10.3390/vaccines10081271

**Published:** 2022-08-07

**Authors:** Muhammad Chutiyami, Dauda Salihu, Umar Muhammad Bello, Stanley John Winser, Amina Abdullahi Gambo, Hadiza Sabo, Adam Mustapha Kolo, Hussaina Abubakar Jalo, Abdullahi Salisu Muhammad, Fatima Ado Mahmud, Khadijat Kofoworola Adeleye, Onyinye Mary Azubuike, Ibitoye Mary Bukola, Priya Kannan

**Affiliations:** 1School of Nursing and Midwifery, Faculty of Health, University of Technology Sydney, Sydney 2007, Australia; 2School of Nursing, The Hong Kong Polytechnic University, Hong Kong 999077, China; 3College of Nursing, Jouf University, Sakaka 42421, Saudi Arabia; 4Department of Physiotherapy and Paramedicine, School of Health and Life Sciences, Glasgow Caledonian University, Glasgow G4 0BA, UK; 5Department of Rehabilitation Science, The Hong Kong Polytechnic University, Hong Kong 999077, China; 6Department of Paediatrics, Aminu Kano Teaching Hospital (AKTH), Kano 700101, Nigeria; 7School of Basic Midwifery, Shehu Sule College of Nursing and Midwifery Damaturu, Damaturu 100101, Nigeria; 8Family Health International (FHI 360) Borno State, Maiduguri 600215, Nigeria; 9Department of Paediatrics, Yobe State Specialist Hospital, Damaturu 620241, Nigeria; 10Physiotherapy Department, Yobe State University Teaching Hospital (YSUTH), Damaturu 620261, Nigeria; 11Physiotherapy Department, Yobe State Specialist Hospital, Damaturu 620241, Nigeria; 12College of Nursing, University of Massachusetts, Amherst, MA 01003, USA; 13Department of Nursing, Federal Medical Centre Asaba, Isieke Asaba 320213, Nigeria; 14Department of Nursing Science, University of Ilorin, Ilorin 240003, Nigeria

**Keywords:** COVID-19, fear of coronavirus, vaccine uptake, predictors, Nigeria

## Abstract

This study examined the association between COVID-19 and fear of contracting COVID-19 and reasons for vaccination refusal. A population-based online survey was conducted via social media in Nigeria using the Fear of COVID-19 scale and items related to vaccination refusal/hesitancy items. Individuals aged 13 years and older were invited to participate. Data were analysed using binary logistic regression to calculate odds ratios (ORs) and associated 95% confidence intervals (CIs) at a *p*-value of less than 0.05. The study enrolled 577 individuals with a mean age of 31.86 years, 70% of whom were male and 27.7% of whom had received at least one dose of the vaccine against COVID-19. None of the variables on the Fear of COVID-19 scale significantly predicted vaccine uptake in multivariate analysis. However, individuals who were fearful of COVID-19 were more likely to be vaccinated in bivariate analysis (OR: 1.7, 95% CI: 1.06–2.63). The most significant factors among the vaccination refusal items associated with COVID-19 vaccination were doubts about vaccination (adjusted OR: 2.56, 95% CI: 1.57–4.17) and misconceptions about vaccine safety/efficacy (adjusted OR: 2.15, 95% CI: 1.24–3.71). These results suggest that uptake of the vaccine against COVID-19 in Nigeria can be predicted by factors associated with vaccination refusal, but not by fear of COVID-19. To contain the pandemic COVID-19 in Nigeria, efforts should be made to educate people about the efficacy of the vaccine and to increase their confidence in vaccination.

## 1. Introduction

The coronavirus pandemic 2019 (COVID-19), caused by severe acute respiratory syndrome coronavirus 2 (SARS-CoV-2), has caused unprecedented concern around the world, and the number of cases continues to increase worldwide despite efforts to contain the outbreak [1]. The evolution of viral variants has made containment extremely difficult. The spread of the disease continues with uncertainty [2], and some of the emerging SARS-CoV-2 variants have been shown to be highly infectious [3], raising concerns about the efficacy of COVID-19 vaccines [4] and, therefore, requiring further research. 

Vaccines provide active immunity against infectious diseases [5], and the introduction of vaccines has led to the eradication of several diseases in some countries, including smallpox [6] and poliomyelitis [3,7]. Since the beginning of the COVID-19 pandemic, efforts have been made to develop vaccines, and vaccines have been introduced by AstraZeneca, Pfizer, and Moderna, among others [8]. High-income countries, such as Australia, and middle-income countries, such as South Africa, have made considerable progress in national COVID-19 vaccine coverage; however, countries in sub-Saharan Africa have some of the lowest vaccination rates in the world [9].

In Africa as a whole, and Nigeria in particular, vaccine hesitancy remains a concern [10], which may explain the high incidence of several preventable diseases in the country, including poliomyelitis and measles [11]. The increasing number of COVID-19 cases in the country is coupled with widespread vaccination hesitancy, which highlights the importance of assessing COVID-19 vaccination coverage in Nigeria and identifying factors that both promote and discourage vaccination coverage. In addition, there is limited research data on COVID-19 vaccination coverage in Nigeria. Therefore, this study aimed to investigate two areas. First is the relationship between fear of COVID-19 and COVID-19 vaccine uptake. Second, the reasons for vaccine refusal as determinants of COVID-19 vaccine uptake were studied.

## 2. Methods

### 2.1. Study Design

A population-based online survey was conducted to explore the experience of COVID-19 vaccination in the general population of Nigeria. The study followed the Strengthening the Reporting of Observational Studies in Epidemiology (STROBE) guidelines for observational studies [12].

### 2.2. Participants

Participants in this online survey were male and female Nigerians aged 13 years and older. The survey was open to all persons in the six geopolitical zones in Nigeria, including the North-Central, North-East, North-West, South-East, South-South, and South-West regions.

### 2.3. Instruments

Two instruments, the Fear of COVID-19 Scale [13] and the vaccination refusal/hesitancy items [14], were used in this study. The Fear of COVID-19 Scale (Appendix A) is a 7-item questionnaire that contains various questions about different aspects related to fear of coronavirus disease. Each item of the outcome measure is answered on a 5-point scale: strongly disagree, disagree, neutral, agree, and strongly agree. Reliability coefficients were also found to be acceptable (α = 0.72, intraclass correlation coefficient = 0.72) [13]. The self-developed items on vaccination refusal (Appendix A) were taken from the study by Kayode, Babatunde, Adekunle, Igbalajobi and Abiodun [14] and consisted of six items that addressed reasons for vaccination refusal, particularly in the Nigerian context. In addition, the questionnaire was designed to include the demographic data of the target population. Subjects were required to answer “yes” or “no” to each item on the vaccination refusal scale. Both scales were completed in English.

### 2.4. Data Collection

Data for this survey were collected online through social media platforms, including Facebook and LinkedIn. Facebook was selected because it is one of the most popular social media platforms in Nigeria, and LinkedIn was selected because of its linkage to professional networks. The instruments were presented using Google Forms, and the link was shared across social media platforms from October to December 2021 by the first three authors (MC, DS, UMB). There were no participation restrictions throughout the survey period, and the survey link was deactivated on 31 December 2021. COVID-19 vaccination was defined as the receipt of at least one dose of the COVID-19 vaccine at any time since the start of the COVID-19 outbreak in 2019.

### 2.5. Data Analysis

Frequency distribution tables and bar graphs are used to summarise participants’ responses to the online survey. Binary logistic regression was used to examine the individual variables predicting COVID-19 vaccine uptake. Multiple logistic regression (MLR) was used to examine the interaction of factors influencing COVID-19 vaccination. The variables captured by the Fear of COVID-19 scale and the vaccine refusal or hesitancy items were used to create two separate MLR models using the “enter” input strategy. Each model accounted for the effects of three demographic characteristics: Age, sex, and geographic region. The individual variables of the two scales were first analysed at the bivariate level before being included in the multivariate models. Crude and adjusted odds ratios (ORs) were used to evaluate effect sizes for the bivariate and MLR analyses. The alpha level was set at 0.05 with a 95% confidence interval (CI) for all outcomes. Data were coded and analysed using the SPSS statistical package (IBM Corp. IBM SPSS Statistics for Macintosh, version 25.0. Armonk, NY, USA).

### 2.6. Ethical Considerations and Participant Recruitment

Formal ethical approval for this study was obtained from Shehu Sule College of Nursing and Midwifery Damaturu, a higher educational institution in Nigeria (Ref: SSCON&M/RC/EC/01/004). All participants were given a detailed information sheet to read before giving consent to participate in the study (Appendix A). Only participants who agreed and signed their consent electronically were given access to the questionnaire. Participants had the option to skip questions they did not want to answer or to withdraw from the survey at any time without penalty. To protect the privacy of all participants, no identifiable information was requested or used in the analysis of study results. However, participants were given the option to provide an email address at the end of the survey to which a summary of the study results was sent.

## 3. Results

### 3.1. Participant Sociodemographic Characteristics and COVID-19 Vaccination Status

A total of 577 participants with an average age of 31.86 years participated in this study, of which approximately 70% were male. The majority of the participants (62.1%) were from the north-eastern region of Nigeria. Only 27.7% of the respondents had received at least one dose of the COVID-19 vaccine (Table 1).

### 3.2. Fear of Coronavirus Disease and Vaccine Hesitancy among Participants

About half of all participants agreed that they were afraid of coronavirus as a whole (51.9%), afraid of losing their life to coronavirus infection (48.5%), or uncomfortable thinking about coronavirus or seeing news about coronavirus on social media (49.5%). Only 14% reported sleep problems due to worries about coronavirus (Figure 1).

More than 50% of participants expressed doubts about the need for COVID-19 vaccination or about the safety of the vaccine, fear of adverse effects or immune system burden, and misconceptions about the vaccine’s effectiveness. Only 22.9% had had negative experiences with vaccines in the past (Figure 2).

### 3.3. Fear of Coronavirus Disease and COVID-19 Vaccine Uptake

Of the 571 participants who reported their COVID-19 vaccination status, only 507 provided complete information on fear of coronavirus disease; therefore, only 507 responses were included in this regression analysis. Responses to the Fear of COVID-19 scale (Table 2) showed that participants who were fearful of coronavirus disease were twice as likely to have received at least one dose of COVID-19 vaccine in the bivariate analysis (OR:1.7, 95% CI: 1.06–2.63). However, this association did not remain significant when adjusted for age, sex, and geographic region in multivariate analysis (Table 2).

### 3.4. Reasons for Vaccine Refusal and COVID-19 Vaccine Uptake

Of the 571 participants who reported their COVID-19 vaccination status, only 505 provided complete information on reasons for vaccination refusal; therefore, only 505 responses were included in this regression analysis. The items on vaccination refusal/hesitancy items (Table 3) revealed that fear of adverse events, concern about immune system exposure, and misconceptions about vaccine safety/efficacy significantly predicted vaccination uptake. After controlling for the effects of age, sex, and geographic region, doubts about the need for vaccination, concerns about immune system exposure, and misconceptions about vaccine safety/efficacy significantly predicted vaccination uptake. However, individuals who had no negative past experience with vaccines were less likely to be vaccinated in both bivariate and multivariate logistic regressions (Table 3).

## 4. Discussion

To our knowledge, this is the first study to examine the association between fear of COVID-19, reasons for vaccination refusal, and COVID-19 vaccination in Nigeria. Overall, fear of coronavirus disease did not significantly predict COVID-19 vaccine uptake, but several reasons for vaccine refusal significantly predicted COVID-19 vaccine uptake, including concerns about vaccine safety and efficacy, previous negative vaccine experiences, and immune exposure concerns. After controlling for the effects of demographic variables, these factors, along with doubts about the need for vaccination, predicted COVID-19 vaccine uptake.

Consistent with previous studies, a positive correlation has been found between vaccine acceptance, health-related fears, and vaccination uptake [15]. The literature has shown that vaccine refusal is related to doubts about the technologies used for vaccine production, personal safety concerns, doubts about vaccine efficacy, lack of knowledge about the development of immunity, belief in COVID-19 misinformation about vaccines, negative past experiences with vaccines, and vaccine-specific concerns, including poor quality or toxicity of vaccine components and the technology used for vaccine production [16,17,18,19,20,21]. In contrast, concern about the risks of infection, previous positive vaccination experiences, and a low belief that the vaccine could lead to side effects have been found to influence vaccine acceptance [16,21]. Equally important, previous research has shown that trust in authorities, desire to build immunity using vaccines, belief in vaccine efficacy/safety, and improved access to COVID-19 vaccine information predict vaccine acceptance [22,23,24]. Consistent with our findings, belief in conspiratorial narratives or misinformation about vaccine safety, negative past vaccination experiences, and doubts about vaccination have been found to contribute to vaccination refusal or hesitation [16,25].

The literature suggests that vaccine refusal and acceptance are influenced by the interplay of many factors, ranging from COVID-19 safety concerns to previous community experience [26]. However, a key prerequisite for vaccine acceptance is trust in government and expert institutions [27,28]. Trust in government has already been found to influence vaccine acceptance, while lack of trust is a major factor in vaccine rejection [29,30,31]. Public awareness is also important. According to innovation diffusion theory, awareness is a step toward acceptance of new technological innovation [32]. In the medical literature, awareness is central to human behaviour and promotes understanding of impending needs [33]. Therefore, the Nigerian public may not have been fully informed about some aspects of vaccine development (including safety and efficacy), leading to increased vaccine refusal. Vaccine safety is another factor that people consider when deciding whether to vaccinate [28]. A vaccine that has been proven safe and effective by the government is more likely to be accepted [16]. Concerns about vaccine safety influence COVID-19 vaccine acceptance and rejection; in particular, individuals with a strong belief in vaccine safety and efficacy are more likely to accept vaccination [21,34], whereas individuals who view the vaccine development process as rushed, potentially compromising efficacy and safety, are less likely to trust and more likely to reject the vaccine [35,36]. Government policies may also affect vaccine uptakes, such as public access requirements, international travel, and employer regulations.

The potentially harmful spread of misinformation about vaccine safety could compromise health care and endanger public health [37]. Negative misinformation about vaccines could also be responsible for lower vaccination rates in communities, as negative misinformation leads to uncertainty, confusion, and mistrust [38]. People who are less fearful of COVID-19 may also have misinformation about the potential threat COVID-19 poses to their health and that of their families [39]. Adequate vaccination coverage is considered a necessary component of returning to pre-pandemic normalcy in the context of coronavirus disease [40], and increased awareness of the threat posed by COVID-19 correlates with higher vaccination acceptance [37]. Perceived risks and associated fear of coronavirus infection have been shown to influence people’s health behaviours [41] and were positively related to willingness to be vaccinated with the COVID-19 vaccine [39]. The observed COVID-19 vaccination refusal in the current study cohort may be due to the perceived impact of racial profiling spread via lethal misinformation on social media, including the notion that COVID-19 does not affect the black population [42,43] or is less severe due to the influences of tropical diseases such as malaria [44], despite evidence that no-one is immune to the effects of COVID-19. Prior experience is another predictor of vaccine acceptance or rejection; in particular, persons with prior SARS-CoV-2 infection are more likely to accept the vaccine [45], whereas persons with negative vaccination experience are more likely to reject or be reluctant to accept the vaccine. 

Concerns about immune exposure were identified as a factor influencing vaccine acceptance, and those who were unwilling to accept the COVID-19 vaccine expressed concerns about genetic alterations, concerns about the short duration of immunity, concerns about the possible toxicity of the vaccine, concerns the poor quality of vaccine components, concerns about the novelty of the antigen delivery system, perception of the disease as a biological weapon (unnatural origin), doubts about the technology used in vaccine development, belief that microchips could be injected into subjects through the vaccine, or fears that the vaccine could lead to infertility [17,19,20,25,46,47]. Individuals who perceived COVID-19 as a naturally occurring disease that is more deadly and contagious than the H1N1 influenza virus and were aware of the possibility of another wave of infection were more likely to accept the vaccine [48].

People have also been shown to be reluctant to accept COVID-19 vaccines for fear of adverse reactions [19,35]. Accordingly, this study found that individuals who are not afraid of vaccine side effects or have no concerns about vaccine safety/efficacy are twice as likely to accept the COVID-19 vaccine (Table 3). These participants may consider the disease to be high-risk or consider side effects to be less severe. One systematic review concluded that adverse events associated with vaccines are extremely rare [49] and that people may perceive lower risks associated with vaccines if they acknowledge that vaccines rarely result in very serious adverse reactions [50]. Similarly, positive attitudes toward vaccination may increase vaccine acceptance, and those who were willing to accept the COVID-19 vaccine perceived its positive effects and were anxious to protect themselves and their loved ones [51]. These behaviours can be further explained by reasoned action theory, which emphasises the importance of beliefs in behavioural predictions, and suggests that individual behavioural intentions can be predicted by attitudes toward behavioural outcomes and social environmental norms [52].

### 4.1. Limitations

The generalisability of the study results to the general Nigerian population may be affected by low response rates among some subgroups, particularly among participants in the southern regions of the country; therefore, our study is associated with a lack of national representativeness. Response rates were generally low relative to the overall population, which could be due to poor access to Internet services. These results are also difficult to generalise to the international community. Another limitation was the use of only two scales, the Fear of COVID-19 scale and the vaccine refusal/hesitancy items, without consideration of socioeconomic and other potentially influencing factors. Research has shown that fear of COVID-19 is multifaceted and can include socioeconomic challenges and stress-related symptoms (e.g., nightmares) [45,53]. Therefore, other indirect aspects of fear may have correlated with the willingness to vaccinate against COVID-19. Further studies should be conducted to examine the influence of the other factors associated with COVID -19 fear on COVID-19 vaccine acceptance. Our results on vaccine refusal/fear of COVID-19 should be taken with caution because the vaccine refusal instrument we used is a self-developed instrument that has not been tested for validity and reliability. In addition, the sensitivity and specificity of the Fear of COVID-19 scale have yet to be confirmed.

### 4.2. Implications

Governments, policymakers, and other health care stakeholders should consider the findings of this study when designing campaigns to promote vaccine acceptance. Further studies should examine the influences of socioeconomic and stress factors on willingness to accept the COVID-19 vaccine. To achieve an adequate response rate for this type of national survey, a hybrid survey format (in-person and online) is recommended. 

## 5. Conclusions

This study examined the association between fear of COVID-19, factors associated with vaccination refusal, and COVID-19 vaccination uptake. Our results showed that doubts about the need for vaccination, concerns about immune exposure, previous negative vaccination experiences, and misconceptions about vaccine safety and efficacy were factors that militated against COVID-19 vaccination. The study, therefore, concludes that COVID-19 vaccination coverage can be predicted by factors related to vaccination refusal, but also by factors related to fear of coronavirus disease in Nigeria.

## Figures and Tables

**Figure 1 vaccines-10-01271-f001:**
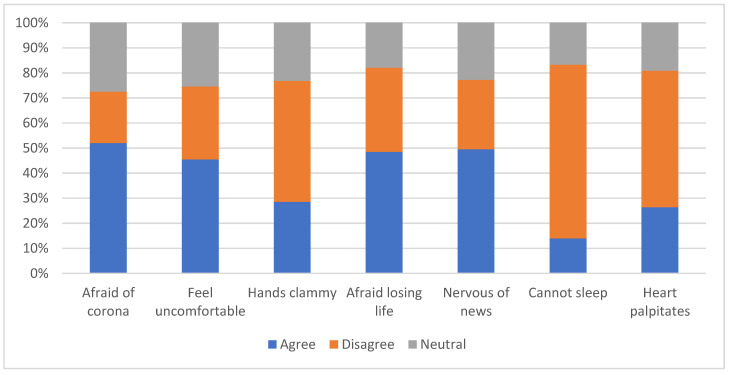
Multidimensional bar chart indicating fears associated with COVID-19 assessed by the Fear of COVID-19 Scale.

**Figure 2 vaccines-10-01271-f002:**
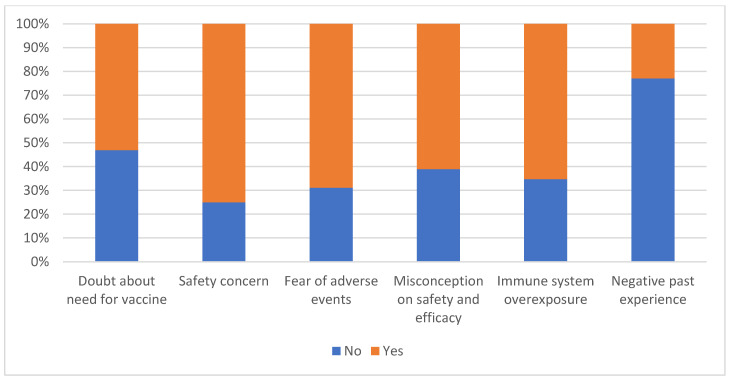
Multidimensional bar chart indicating reasons for vaccine refusal assessed by vaccine refusal/hesitancy items.

**Table 1 vaccines-10-01271-t001:** Participant demographics and COVID-19 vaccination status.

Variables	Categories	Responses	Percentage
Age, y (Mean ± SD)	–	31.86 ± 9.10	–
Sex	Female	180	31.3
	Male	395	68.7
Geopolitical zone	North-east	356	62.1
	North-west	123	21.5
	North-central	42	7.3
	South-east	11	1.9
	South-south	20	3.5
	South-west	21	3.7
COVID-19 vaccination status	Vaccinated	158	27.7
	Unvaccinated	413	72.3

SD, standard deviation.

**Table 2 vaccines-10-01271-t002:** Association between fear of coronavirus disease and COVID-19 vaccine uptake (n = 507).

Variables	Categories	Crude OR (95% CI)	Adjusted OR (95% CI)
I am most afraid of corona	Agree	1.70 (1.06–2.63) *	1.34 (0.72–2.51)
	Disagree	1.40 (0.90–2.45)	1.36 (0.70–2.63)
	Neutral	1	1
It makes me uncomfortable to think about corona	Agree	1.53 (0.96–2.44)	1.48 (0.78–2.78)
	Disagree	1.03 (0.61–1.73)	0.84 (0.43–1.63)
	Neutral	1	1
My hands become clammy when I think of corona	Agree	1.14 (0.68–1.92)	0.73 (0.38–1.42)
	Disagree	1.12 (0.70–1.79)	1.42 (0.77–2.61)
	Neutral	1	1
I am afraid of losing my life because of corona	Agree	1.24 (0.75–2.07)	1.17 (0.59–2.32)
	Disagree	0.86 (0.49–1.49)	0.98 (0.50–1.96)
	Neutral	1	1
When I watch news about corona on social media, I become nervous	Agree	1.26 (0.78–2.03)	0.95 (0.52–1.75)
	Disagree	1.17 (0.69–1.98)	1.42 (0.75–2.69)
	Neutral		1
I can’t sleep because I am worried about getting corona	Agree	1.17 (0.62–2.24)	0.96 (0.42–2.22)
	Disagree	0.86 (0.52–1.41)	0.99 (0.51–1.90)
	Neutral	1	1
My heart palpitates when I think about getting corona	Agree	0.98 (0.58–1.68)	0.85 (0.42–1.69)
	Disagree	0.63 (0.40–1.04)	0.47 (0.25–0.90)
	Neutral	1	1
Sex	Male	0.79 (0.53–1.16)	0.88 (0.55–1.40)
	Female	1	1
Geopolitical zone	North-east	1.00 (0.38–2.65)	0.79 (0.27–2.30)
	North-west	0.77 (0.27–2.17)	0.75 (0.24–2.31)
	North-central	0.78 (0.24–2.55)	0.50 (0.14–1.82)
	South-east	1.43 (0.30–6.74)	0.82 (0.13–5.12)
	South-south	1.68 (0.45–6.13)	1.02 (0.99–1.04)
	South-west	1	1
Age	–	1.02 (1.00–1.04)	1.02 (0.99–1.04)

* *p* < 0.05; OR, odds ratio; CI, confidence interval.

**Table 3 vaccines-10-01271-t003:** Association between reasons for vaccine refusal and COVID-19 vaccination uptake (N = 505).

Variables	Categories	Crude OR (95% CI)	Adjusted OR (95% CI)
Doubt about vaccination	No	2.79 (1.90–4.09)	2.56 (1.57–4.17) ***
	Yes	1	1
Concerns about vaccine safety	No	1.26 (0.83–1.91)	0.72 (0.43–1.24)
	Yes	1	1
Fear of adverse events	No	2.14 (1.46–3.15) ***	0.94 (0.53–1.68)
	Yes	1	1
Misconceptions about safety and efficacy	No	2.92 (2.00–4.26) ***	2.15 (1.24–3.71) **
	Yes	1	1
Concerns about immune system exposure	No	2.25 (1.54–3.29) ***	1.79 (1.06–3.01) *
	Yes	1	1
Negative past experience with vaccine	No	0.62 (0.41–0.95) *	0.54 (0.33–0.89) *
	Yes	1	1
Sex	Male	0.79 (0.53–1.16)	0.71 (0.44–1.16)
	Female	1	1
Geopolitical zone	North-east	1.00 (0.38–2.65)	1.14 (0.38–3.38)
	North-west	0.77 (0.27–2.17)	0.80 (0.26–2.51)
	North-central	0.78 (0.24–2.55)	0.72 (0.20–2.67)
	South-east	1.43 (0.30–6.74)	1.14 (0.17–7.72)
	South-south	1.68 (0.45–6.13)	0.91 (0.19–4.28)
	South-west	1	1
Age	–	1.02 (1.00–1.04)	1.03 (1.00–1.05)*

* *p* < 0.05, ** *p* < 0.01, *** *p* < 0.001; OR, odds ratio; CI, confidence interval.

## Data Availability

The data presented in this study are available on request from the corresponding author. The data are not publicly available due to ethical reasons.

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
