# Peer review of "Are Fear of COVID-19 and Vaccine Hesitancy Associated with COVID-19 Vaccine Uptake? A Population-Based Online Survey in Nigeria"

_vaccines, 2022, doi:10.3390/vaccines10081271_

Round 1

Reviewer 1 Report

Thank you for the invitation. This study is an interesting piece of work, however; there are several concerns to be addressed here.

1. Authors have used two tools during their study. Though these tools are previously validated in other populations, authors have opted for the same and conducted a study on the Nigerian population. I did not see any information on how these two tools are validated, what were the mechanisms of the translation, and how the reliability of the tool was done. Please provide this information in the manuscript. Without validation, the study may have erroneous results which may arise due to the non-understandability of the study too.

Please provide the tool (English and translated version) in the supplementary file of the manuscript.

Table 2: what does COVID-19 vaccine uptake mean? authors have listed that 507 intended to uptake the vaccine but there is no information in the results that how these numbers calculated? is it intention to take the vaccine? if so then how such intention was estimated, as there is no information in the manuscript about such estimation. In table 3, these numbers are 505 which is showing that vaccine uptake was estimated by two different methods? Here the confusion will arise among readers as the methodology section does not provide definitions that what is meant by vaccine uptake, vaccine refusal, vaccine acceptance, vaccine hesitancy etc..

It is also important to compare the vaccine hesitancy and fear between vaccinated and non-vaccinated respondents. By this way, clearer picture will come to show the perception of people to get the vaccine, and this analysis will also identify the more important factor involved in vaccine refusal among non-vaccinated candidates.

How many respondents who were vaccinated still reluctant, hesitant and have phobia towards the vaccine?

Authors may consider doing relative importance index analysis as done by some papers to rank the factors associated with vaccine hesitancy, acceptance or refusal (https://www.ncbi.nlm.nih.gov/pmc/articles/PMC5893193/).

Reviewer 2 Report

1. Research design and methods 

2 separate instruments were used in the study (i) fear of COVID-19 scale (Ahorsu et al, 2020) (ii) vaccine refusal/ hesitancy items (Kayode et al, 2021). The fear of COVID-19 scale was developed from a convenience sample of an Iranian population with several limitations including the sensitivity and specificity of the scale could not be examined because the study participants were from the general population and verification was needed using confimatory factor analysis. The reference for the second instrument was a letter to the Editor with a short descriptions of the causes of vaccine hesitancy. No survey instruments was presented in the paper. Vaccine hesitancy is in an interplay of many factors including health related fears and risks of infections. How the 2 sets of items from different sources were combined and validated needs to be clarified and justified.

2. 2 separate multiple logistic regression models from 2 sets of items from different sources were constructed and the items relating t ‘fear of COVID 19’ and ‘vaccine hesitancy’ were analysed separately. There is no justification or description on the relationship between these two domains in examining their association with vaccine uptake. The similarity of items in both scales has the potential result in multi-collinearity in multiple regressions. There is no test accounting for this issue. 

3. In Table 3, people without negative past experience with vaccine seems to be less likely to be vaccinated (AOR = 0.54). This outcome is not consistent with descriptions in the text. 

Reviewer 3 Report

This observational study reports data on the association between fear of COVID-19 and COVID-19 vaccine uptake, as well as the reasons for vaccine refusal. The topic is interesting and highlights the importance of pursuing health promotion policies, in particular about vaccination. As also specified in the introduction, in Nigeria and in the sub-Saharan regions vaccine hesitancy remains a concern which may explain the high incidence of various preventable diseases: knowing the causes of vaccine hesitation, it will be possible to promote more efficient vaccination campaigns.

The article is written in clear and exhaustive way. The review is complete in its essential parts. Here are some indications:

Point 1: The Fear of COVID-19 Scale present in Appendix 2 shows five possible answers for each question, unlike what is written in the methods section where it is reported that the possible answers are three. Also in the results section and in the figures only three answers are reported. Have 'strongly disagree' and 'disagree' and then 'strongly agree' and 'agree' been merged? If yes, please specify it in the text.

Point 2: The discussion focuses on many aspects, the bibliography as proof of what is cited is consistent. The reasons for the refusal of the vaccine are listed in the first part and then be explored in the following paragraphs. The discussion is also interesting to learn more about the Nigerian context relating to public health. Nevertheless, I would tend to summarize some concepts, especially in the first part, such as mistrust in institutions and conspiracy theories that, however useful to understand the reasons for vaccine hesitation, are mostly the result of literature research and not directly derived from the study.

Point 3: Please check the text: in some places there are grammatical errors, typos or missing words.

Round 2

Reviewer 1 Report

Authors have addressed the comments.

Author Response

Thank you for requesting the English Language and style spell check. We conducted a language and technical editing of the manuscript.